# Activation of Slit2/Robo1 Signaling Promotes Tumor Metastasis in Colorectal Carcinoma through Activation of the TGF-β/Smads Pathway

**DOI:** 10.3390/cells8060635

**Published:** 2019-06-25

**Authors:** Yuying Yao, Zijun Zhou, Liuyou Li, Junchen Li, Lixun Huang, Jiangchao Li, Cuiling Qi, Lingyun Zheng, Lijing Wang, Qian-Qian Zhang

**Affiliations:** Vascular Biology Research Institute, School of Life Sciences and Biopharmaceutics, Guangdong Pharmaceutical University, Guangzhou 510006, China; yaoyuying93@163.com (Y.Y.); zhouzijun72@163.com (Z.Z.); 18028014631@163.com (L.L.); lijunchen96@163.com (J.L.); gdpuhlx@163.com (L.H.); lijiangchao1234@163.com (J.L.); qicuiling12345@163.com (C.Q.); freeflyzly@hotmail.com (L.Z.)

**Keywords:** colorectal carcinoma, metastasis, Slit2/Robo1 signaling, TGF-β/Smads signaling, R5

## Abstract

Slit2 (slit guidance ligand 2), a ligand of the Roundabout1 (Robo1) transmembrane receptor, is often overexpressed in colorectal carcinomas (CRCs). In this study, we performed data mining in the Metabolic gEne RApid Visualizer (MERAV) database and found that Slit2 and TGF-β1 (Transforming growth factor-β1) are highly expressed in carcinomas relative to those in tumor-free tissues from healthy volunteers or wild type mice. Furthermore, expression of Slit2 and TGF-β1 in CRCs increases with pathological stages. Serum levels of Slit2 in patients with CRC and in Apc^Min/+^ mice with spontaneous intestinal adenoma were significantly increased compared with those in healthy controls. Specific blockage of Slit2 binding to Robo1 inactivated TGF-β/Smads signaling and inhibited tumor cell migration and metastasis, which can be partially restored by treatment with TGF-β1. However, specific inhibition of TGF-β1/Smads signaling reduced CRC tumor cell migration and invasion without affecting cell proliferation. This study suggests that activation of Slit2/Robo1 signaling in CRC induces tumor metastasis partially through activation of the TGF-β/Smads pathway.

## 1. Introduction

Colorectal carcinoma (CRC) is one of the most common types of cancer worldwide. The death rate of CRC without metastasis has been falling in recent years. However, distant metastasis is the major cause of cancer death in patients with CRC [1,2,3]. Distance metastasis is often observed in the most advanced stage of CRC, and the therapeutic effects for metastatic patients with CRC are limited, leading to a poor survival rate [4,5]. The molecular mechanism underlying tumor metastasis of CRC is still unclear. Therefore, the development of a biomarker—which could not only be used to diagnose cancer but also as a therapeutic target—is key to the development of a novel therapeutic approach, thereby improving the survival rate of patients with CRC.

Slit glycoprotein (Slit1, 2 and 3) can bind to the Roundabout receptor (Robo1, 2, 3, and 4) and activate Slit/Robo signaling, which plays an important role in the cell migration involved in physiological and pathological processes [6,7,8]. Differences in the expression and function of Slit/Robo signaling have been discovered in various cancers [7,9,10,11]. The promoter’s hypermethylation often leads to Slit2 down-regulation in most types of cancers, including CRC [12,13,14,15]. Additionally, methylation of Slit2 does not associate with the clinicopathological stage in Wilms’ tumor and renal cell carcinoma [16]. Accordingly, our previous studies have shown that Slit2 overexpression occurs in most of patients with CRC, especially in late-stage patients [8,17,18]. Slit2 binds to Robo1, which is mainly expressed in tumor cells, and its important role in the regulation of tumor growth and metastasis in CRC has been extensively studied. However, the molecular mechanisms of Slit2/Robo1 signaling in the regulation of tumorigenesis and cancer progression of CRC still require further exploration.

Transforming growth factor-β (TGF-β) is significantly increased in cancer tissues and the serum in CRC [19,20]. Additionally, the expression of TGF-β is increased in parallel with the tumor’s pathological stage in melanoma [21]. The TGF-β family consists of three isoforms that include TGF-β1, TGF-β2, and TGF-β3. TGF-β1 is the most abundant and well-studied isoform of the TGF-β family members, and plays an important role in the development and progression of CRC [22,23]. Accordingly, high levels of TGF-β1 in the primary tumor of CRC are correlated with progression, recurrence, and a decreased survival rate [24,25]. It has been reported that activation of TGF-β and Wnt signaling in CRC cells can induce the expression of Robo1. However, the relationship between Slit2/Robo1 signaling and the TGF-β1 pathway needs to be further clarified [26].

TGF-β can bind to the TGF-β receptor II (TGF-βRII), and then phosphorylate to TGF-βRI. Activated TGF-βRI phosphorylates the intracellular downstream transcription factors, Smad2 and Smad3, to regulate the transcription of TGF-β-responsive genes [27]. TGF-β is a critical cytokine that has dual responses in cancer. It has been established that the TGF-β pathway acts as a tumor suppressor to suppress tumor formation in early stage tumors [28]. However, TGF-β can also act as an oncogenic factor to induce tumor progression and metastasis in the advanced stages of cancers [28,29,30]. Studies have shown that TGF-β1 can induce epithelial–mesenchymal transition (EMT) in most CRCs, which contributes to the cells becoming of an invasive phenotype [31].

This study further investigates the possible role and the molecular mechanism of Slit2/Robo1 signaling in activating the TGF-β1/Smad2/3 pathway, leading to an induction of the invasive phenotype of CRC cells.

## 2. Materials and Methods

### 2.1. Reagents and Antibodies

R5, a monoclonal antibody that specifically recognizes human, rat and mouse Robo1, and its isotype-matched mIgG (Immunoglobulin G), were prepared as previously reported [18]. P144 (Disitertide, HY-P0118), the TGF-β1 inhibitor, was purchased from MedChem Express (MCE, Monmouth Junction, NJ, USA). Recombinant human TGF-β1 (100–21) was purchased from PeproTech (Rocky Hill, NJ, USA). Rabbit anti-TGF-β1 (BA0290, diluted at 1:100 for IHC (immunohistochemical staining), and 1:250 for WB (western blotting assay)) and rabbit anti-p-Smad3 (BM4033, diluted at 1:50 for IHC) were purchased from Boster (Wuhan, China); rabbit anti-Smad2/3 (sc-376928, diluted at 1:500 for IHC) were purchased from Santa Cruz Biotechnology (Dallas, TX, USA); rabbit anti-p-Smad2 (abs131074, diluted at 1:50 for IHC) were purchased from Absin (Shanghai, China); rabbit anti-Robo1 (ab7279, diluted at 1:1000 for WB) was purchased from Abcam (Cambridge, UK); and rabbit anti-Smad2/3 (8685T, diluted at 1:1000 for WB), rabbit anti-p-Smad2 (3108T, diluted at 1:1000 for WB), rabbit anti-p-Smad3 (9520T, diluted at 1:1000 for WB), and rabbit anti-GAPDH (Glyceraldehyde-3-phosphate Dehydrogenase, 2118, diluted at 1:5000 for WB) were purchased from Cell Signaling Technology, Inc. (Danvers, MA, USA).

### 2.2. Human Serum Samples and Treatment

Serum samples were taken from 23 patients with clinically diagnosed colorectal carcinoma, and 18 healthy volunteers that were used as controls. All clinical serum samples were collected from the first affiliated hospital of Guangdong Pharmaceutical University (Guangzhou, China), and stored at −80 °C until used to analyze the level of Slit2 by ELISA (enzyme linked immunosorbent assay, Cusabio Biothech Co., Wuhan, China) according to the manufacturer’s instructions. Written informed consent was obtained for each participant, and the study was in accordance with the Declaration of Helsinki and the guidelines approved by the Clinical Ethics Committee of the first affiliated hospital of Guangdong Pharmaceutical University (permit No.: 2014(13)).

### 2.3. Animals and Treatment

Apc^Min/+^ mice (C57BL/6J background) were obtained from the Jackson Laboratory (Bar. Harbor, ME, USA). C57BL/6J mice (C57) and BALB/c athymic nude mice (5 weeks old, male) were obtained from the Guangdong Medical Laboratory Animal Center. All mice were maintained in a 12 h light/dark cycle, in a controlled constant temperature and humidity room at 24 ± 2 °C and 60 ± 5% humidity. Use and experiments on the animals in this project were in accordance with the protocols approved by the Center of Laboratory Animals Ethics Committee of Guangdong Pharmaceutical University (permit No.: SYXK(YUE)-2012-0125) and the “Guide for the Care and Use of Laboratory Animals” by the National Academy of Sciences (NIH publications No. 80-23, revised 1996). In addition, all possible efforts were made to minimize suffering.

Lovo human colorectal carcinoma cells (10^6^ cells were suspended in 200 μL PBS (phosphate buffer saline)) were injected into BALB/c athymic nude mice through the caudal tail vein to construct a lung metastasis mouse model according to the previously reported method and the mice were randomly divided into two groups [32]. At the next day after injection, the mice were intraperitoneally injected with mIgG or R5 (a monoclonal antibody that specifically recognizes human, rat, and mouse Robo1) every two days for three weeks using the dose of 1 mg per injection. After the treatment, the mice were sacrificed and the lung metastatic foci were identified.

### 2.4. Cell Culture

Human colorectal carcinoma cell lines, Lovo (highly invasive) and SW480 (poorly invasive), which contain a high expression of Slit2 [8,33], were kindly provided by the Stem Cell Bank, Chinese Academy of Sciences, and cultured in RPIM 1640 (Gibco, Grand Island, NY, USA) and Dulbecco’s modified Eagle’s medium (DMEM, Gibco) with high glucose, respectively, supplemented with 100 U/mL penicillin, 100 μg/mL streptomycin (Gibco), and 10% fetal bovine serum (FBS, Gibco). Then, the cells were maintained in a humidified chamber with 5% CO_2_ at 37 °C.

### 2.5. Data Collection and Analysis

The data used in this study were obtained from the Metabolic gEne RApid Visualizer database (MERAV, http://merav.wi.mit.edu). The datasets used for analysis of the expression of Slit2 and TGF-β1, both in normal tissues and in primary cancer tissues, came from a total of 19 normal and 321 cancerous tissue samples of CRC. The cancer tissues of CRC in the datasets (including 13 grade 1, 241 grade 2, 63 grade 3, and 4 grade 4) were used to analyze the relationship between Slit2 or TGF-β1 expression in CRC, and its clinical pathological stage.

### 2.6. Wound Healing Assay

Lovo cells were seeded in a 12-well plate (2 × 10^5^ cells/well), and a scratch was created with a pipette tip in the middle of the well when the cells reached confluence. Then, the wells were washed with phosphate buffer saline (PBS) to remove non-adherent cells. Drugs were added into the wells, and then the cells were photographed at 0 h and 24 h after drug treatment. The migrated distances of cells from the wound edge were measured from 0 h.

### 2.7. Transwell Migration Assay

The Lovo cells (10^5^ cells) mixed with mIgG or R5 in serum free DMEM were added into the upper of the transwell chambers (8 μm pores, Costar, Cambridge, MA, USA), and then placed in a 24-well plate that included 600 μL of DMEM with 10% of serum. Following 16 h of incubation, the membranes of the chambers were fixed with methanol and stained with 1% crystal violet. The number of cells that migrated to the lower membranes was counted.

### 2.8. Matrigel Invasion Assay

The transwell chambers with 8 μm pores (Costar, Cambridge, MA, USA) were coated with 50 μL of a mixture of serum free DMEM and Matrigel (1:5 dilution, BD Biosciences, CA, USA), and then placed in a 24-well plate for 7 h at 37 °C. After solidification, the chambers were placed in the wells of a 24-well plate, including 600 μL of DMEM with 10% of serum, and the Lovo cells (1.5 × 10^5^ cells) mixed with mIgG or R5 in serum free DMEM were added into the upper chambers. Following 20 h of incubation, the membranes of the chambers were fixed with methanol and stained with 1% crystal violet. Then, the cells on the upper membranes of the chambers were wiped, and the number of cells that had invaded the membranes was counted.

### 2.9. Histological and Immunohistological Analyses

The intestinal tissues were fixed in 10% neutral formalin solution and embedded in paraffin. The sections (4 μm in thickness) were stained with hematoxylin and eosin (H&E) for routine histology examination. For the immunohistochemistry assay, the paraffin sections (4 μm in thickness) were deparaffinized in xylene, dehydrated in a graded series of alcohol, and blocked in 10% bovine serum albumin (BSA) at 37 °C for 30 min. Then, the sections were incubated with relevant primary antibodies and horseradish peroxidase (HRP)-conjugated secondary antibodies. Finally, the sections were incubated with diaminobenzidine (DAB) to develop the signals, and then counterstained with hematoxylin. All images of the sections were captured with a microscope. The protein expression levels in slices were quantified using IPP software (Image-Pro Plus 4.5 software, Media Cybernetics, Silver Spring, MD, USA).

### 2.10. Western Blotting Assay

The total proteins of the cells were extracted using RIPA buffer, and separated using 10% (*w*/*v*) Sodium dodecylsulphate-polyacrylamide gel electrophoresis (SDS-PAGE) before being transferred onto a polyvinylidene fluoride (PVDF) membrane (Millipore, MA, USA). The membranes were blocked using 5% (*w*/*v*) non-fat powder milk for 1 h at room temperature, and incubated in the primary antibody overnight at 4 °C. Then, the membranes were incubated with the relative horseradish peroxidase (HRP)-conjugated secondary antibodies at room temperature for 1 h, and further visualized by exposure to the Image Quant LAS 4000 system (GE Healthcare, Waukesha, WI, USA). GAPDH was used as the loading control. The protein bands were quantified densitometrically using Quantity-One protein analysis software (Bio-Rad Laboratories, Hercules, CA, USA), which normalized the GAPDH expression.

### 2.11. Statistical Analysis

All data are represented as the mean ± standard deviation (SD) of three independent experiments. Statistical differences between two groups were analyzed using Student’s two-sided *t*-tests. *p* < 0.05 was considered statistically significant.

## 3. Results

### 3.1. Slit2 is Overexpressed in CRC

The MERAV database, which is a collection of aggregate array data for deeper analysis of gene expression in cancers, is used to analyze Slit2 expression in tumor tissues of CRC [34]. The expression of Slit2 was significantly increased in the tumor tissues of CRC patients compared to normal human intestine tissues (Figure 1A). In addition, Slit2 expression in the tumor tissues of patients with CRC was gradually upregulated with the increase of the CRC pathological stage, and was markedly higher than that in normal human intestine tissues when the clinical pathological stage reached stage 2 (Figure 1B).

Next, we investigated the serum levels of Slit2 in 23 patients with CRC and 18 healthy control persons. The serum levels of Slit2 in CRC patients were significantly higher than that in healthy persons (Figure 1C). In addition, the Apc^Min/+^ spontaneous intestinal adenoma mice were employed to further confirm the serum levels of Slit2 during intestinal tumor development. As shown in Figure 1D, the serum levels of Slit2 in Apc^Min/+^ mice were significantly higher than in wild type mice (C57). Moreover, the serum levels of Slit2 gradually increased with tumor development in Apc^Min/+^ mice (Figure 1D and Appendix A). These results suggested that Slit2 signaling is activated during intestinal tumor development and might be involved in the development of pathological processes in CRC.

### 3.2. Specific Blocking of Slit2/Robo1 Signaling Inhibits Tumor Growth and Metastasis of CRC In Vivo

Slit2 binds to Robo1 in tumor cells, and plays an important role in the regulation of tumorigenesis and metastasis [35]. Therefore, the Apc^Min/+^ spontaneous intestinal adenoma mouse model and Lovo cell lung metastasis mouse model were used to further investigate the role of Slit2/Robo1 signaling in the development of CRC. R5, a monoclonal antibody that specifically recognizes human, rat, and mouse Robo1, could inhibit Robo1 expression [8,18]. The tumors in the intestine were categorized as microadenoma (<2 mm) and adenoma (≥2 mm) based on the tumor diameters according to a previous report, and the number of microadenomas were higher than that of adenomas in Apc^Min/+^ mice [36]. We found that the tumor incidence (number of tumors) and tumor burden (total volume of tumors per mouse) of microadenoma were significantly inhibited in R5-treated Apc^Min/+^ mice compared with mIgG-treated mice (Figure 2A,B). Therefore, the tumor number and tumor burden were also decreased in adenoma. Unexpectedly, no statistical differences in adenoma between R5- and mIgG-treated Apc^Min/+^ mice were found due to remarkable differences among individuals. Meanwhile, the pathological process of the tumor was observed and it was found that the pathological stage of tumors in R5-treated Apc^Min/+^ mice was decreased compared with the mIgG group (Figure 2C).

Furthermore, we used R5 (1 mg per injection, every two days for three weeks) to treat Lovo cell lung metastasis model mice and found that R5 could markedly inhibit metastatic foci on the surface of lung tissues compared with those treated with mIgG (Figure 2A). In addition, a histologic analysis was carried out and the results showed that R5 decreased the number of the micrometastatic foci in the slices compared to treatment with mIgG (Figure 2B,C). All the results suggested that the specific blocking of Slit2/Robo1 signaling inhibited tumor growth and metastasis during intestinal tumor development.

### 3.3. Blocking Slit2/Robo1 Signaling Suppresses Cell Growth, Migration, and Invasion In Vitro

Lovo cells and SW480 cells were treated with different concentrations of R5. The results showed that specific blocking of Slit2/Robo1 signaling could inhibit cell growth in a concentration-dependent manner in Lovo cells (Figure 3A) and SW480 cells (Appendix A). Then, a colony formation experiment was employed to further confirm the inhibition effect of R5 on the cell proliferation of Lovo cells and SW480 cells. The results showed that R5 significantly inhibited the size and number of colonies compared with the mIgG group in Lovo cells (Figure 3B,C) and SW480 cells (Appendix A).

In addition, a wound healing assay and transwell migration assay were conducted to examine the migration and cell chemotactic capacity of Lovo cells and SW480 cells that were treated with R5 or mIgG. R5, but not mIgG, remarkably suppressed cell migration and chemotaxis in Lovo cells (Figure 3D,E) and SW480 cells (Appendix A). Meanwhile, the Matrigel were coated on the membrane of transwell chambers, the cells mixed with mIgG or R5 were added in the upper membrane of the chambers, and were incubated for 20 h to let cell invasion into the lower membrane. The cell invasion ability was also inhibited by R5 compared with mIgG in Lovo cells (Figure 3F) and SW480 cells (Appendix A). These results indicated that blocking Slit2/Robo1 signaling leads to inhibition of the proliferation, migration, chemotactic, and invasion ability of CRC cells.

### 3.4. Activation of TGF-β/Smad Signaling is Related to Overexpression of Slit2 in CRC

The activation of TGF-β/Smad signaling is responsible for the tumor metastasis in CRC [37]. The analysis results of the data (collected from MERAV database) showed that the expression of TGF-β1 in patients with CRC was higher than that in normal human intestine tissues (Figure 4A). The expression of TGF-β1 was also gradually upregulated with the increase in the CRC pathological stage, and was markedly higher than in normal human intestine tissues when the clinical pathological stage reached stage 2 (Figure 4B). Moreover, TGF-β1 expression was positively correlated with Slit2 upregulation in the tumor tissues of CRC (Figure 4C). Our previous report indicated that inhibition of Slit2/Robo1 signaling-regulated hepatic stellate cell inactivation could reduce the phosphorylation of Smad2 and Smad3 in a TGF-β1-independent way [38]. However, the regulatory relationship between Slit2/Robo1 signaling and TGF-β/Smad activation in CRC still needs to be further clarified. The blocking of Slit2/Robo1 signaling by R5 attenuated TGF-β1 phosphorylation of Smad2 and Smad3 in Lovo cells (Figure 4D), SW480 cells (Appendix A), and tumor tissues of Apc^Min+^ mice (Figure 4E) compared with mIgG-treated groups. These results demonstrated that blocking Slit2 binding to Robo1 could inactivate TGF-β/Smad signaling.

### 3.5. TGF-β/Smads Signaling is Involved in Slit2/Robo1-Induced Tumor Metastasis in CRC

Specific blockage of Slit2 binding to Robo1 could inactivate TGF-β/Smads signaling, although whether TGF-β/Smads signaling is involved in Slit2/Robo1-induced tumor growth and metastasis of CRC is still to be demonstrated. P144 is a synthetic peptide that specifically inhibits TGF-β1 and is often used to inactivate TGF-β/Smads signaling [39]. P144 is used to specifically inhibit TGF-β1 in Lovo cells and SW480 cells. After 48 h of P144 treatment, we observed that cell proliferation was not affected by P144 in Lovo cells (Figure 5A) and SW480 cells (Appendix A). However, cell migration and invasion ability were both significantly suppressed by P144 in Lovo cells (Figure 5B,C) and SW480 cells (Appendix A). These results suggest that TGF-β/Smads signaling mainly regulates tumor metastasis, but not tumor growth, in CRC.

In comparison, the inhibition of Smad2 and Smad3 phosphorylation by R5 was absolutely abolished by TGF-β1 treatment (Figure 5D). Unexpectedly, the inhibition efficacy of R5 on the cell migration, chemotactic, and invasion ability of Lovo cells was partially abolished by treatment with TGF-β1 (Figure 5E–G). This implies that TGF-β/Smads may not be the only signaling pathway that is involved in Slit2/Robo1-mediated tumor metastasis in CRC. These results indicate that Slit2 binds to Robo1 induced tumor metastasis partially through the activation of TGF-β/Smads signaling in CRC.

## 4. Discussion

Our study clearly showed an increased expression of Slit2 in the tumor tissues and serum of patients with CRC compared with healthy controls through an analysis of the MERAV database and ELISA assay detection. Accordingly, we also observed that the serum levels of Slit2 in Apc^Min/+^ spontaneous intestinal adenoma mice were significantly increased compared with wild type mice. In addition, the expression levels of Slit2 in tumor tissues and serum were both gradually increased with intestinal tumor development. Importantly, these experimental results are consistent with our previous findings that an increase in Slit2 expression can be seen in human CRC tissue samples, especially in metastatic tissue samples compared with non-metastatic tissue samples [8,17]. We hypothesized that Slit2/Robo1 signaling might be involved in the tumor development of CRC. In this study, we observed that specific blocking Slit2 binding to Robo1 could attenuate tumor growth and metastasis in intestinal tumor mouse models. Our findings further demonstrate that activation of Slit2/Robo1 signaling is involved in tumorigenesis and metastasis of CRC.

Previous reports have shown that Slit2 plays important roles in various cancers, and the regulatory function of Slit2 is different in different cancers. Slit2 is often inactivated in cancers with promoter region CpG (cytosine--phosphate diester—guanine) island hypermethylation and allele loss, including lung, breast, cervical, intestinal, and hepatocellular carcinomas [12,13,14,15]. In addition, Slit2 often promote tumor development in skin cancer, osteosarcoma, and CRC [8,17,40,41]. However, the functional differences of Slit2 in CRC still require further clarification. It has been reported that Slit2 methylation is not associated with the clinic pathological stage [16]. It was also found that Slit2 methylation is an early event in tumorigenesis and might possibly be associated with different stages of tumorigenesis in CRC [16]. Our work observed the serum levels of Slit2 in Apc^Min/+^ mice at the proliferative stage, adenoma stage, and carcinoma stage, and found that the concentration of Slit2 was increased in the serum of all development stages compared with that in wildtype mice. Accordingly, Slit2 expression was also increased in tumor tissues at different stages in patients with CRC, as shown through analysis using the MERAV database. Additionally, Slit2 was gradually increased during tumor development. Therefore, the CpG island hypermethylation of the Slit2 promoter might occur in a small proportion of patients. The differences in genetic alteration might result in the differential expression of Slit2 in CRC. The expression of Slit2 may provide a direction to distinguish between different types of patients with CRC, and suggest the most appropriate treatment for them.

Slit2 binding to Robo plays a critical role in mediating axon guidance during neural development, mediating angiogenesis, and inducing cancer cell migration [9,42,43]. Our previous findings demonstrated that the activation of Slit2/Robo1 signaling can promote tumorigenesis and metastasis of CRC [8,17]. This effect may arise through activation of the Src-mediated Wnt pathway, and Hakai-mediated epithelial–mesenchymal transition (EMT) phenotype. We also observed that the blocking of Slit2/Robo1 signaling could attenuate phosphorylation of Smad2/3 in a TGF-β-independent way in liver fibrosis [38]. All the findings demonstrate that the regulatory mechanism of Slit2/Robo1 signaling in distinct disease or pathological stages might be quite diverse. In our study, we observed that specific blocking of Slit2/Robo1 signaling could lead to inactivation of the TGF-β/Smads pathway in CRC. The TGF-β and Wnt pathways cooperate to activate gene transcription in intestinal cancer [26]. It has been demonstrated that EMT is beneficial for tumor metastasis, and TGF-β/Smads signaling also contributes to EMT in cancer cells [44]. Surprisingly, we clarified that tumor metastasis, but not tumor growth of CRC is dependent on Slit2/Robo1 signaling induced activation of the TGF-β/Smads pathway. According to our previous findings, Slit2/Robo1 signaling might regulate tumor growth by activation of the Wnt pathway, and induce tumor metastasis by TGF-β/Smads signaling and Hakai-mediated EMT during intestinal tumor development [8,17].

Multiple studies have demonstrated that TGF-β1 is highly expressed in tumor tissues, and associated with advanced stages of CRC [45]. We clarified that TGF-β1 is overexpressed in tumor tissues, and gradually increases with the development of CRC, through an analysis of the MERAV database. Moreover, the expression of TGF-β1 was positively correlated with Slit2. We also found that inhibition of Slit2/Robo1 signaling could decrease TGF-β1. Overexpression of Robo1 can be seen in TGF-β and Wnt-induced intestinal tumors [26]. These results suggest that a possible Slit2–Robo1–TGF-β feedback loop might exist in CRC, through which tumor growth and metastasis are regulated.

TGF-β1 may be a director during CRC progression and can induce tumor invasion and metastasis in the late tumor stages [27]. It has also been clarified that TGF-β1 can induce EMT in most CRCs, which contributes to tumor metastasis [31]. In this study, we found that P144, which inactivates TGF-β1/Smad signaling, cannot inhibit cell proliferation within 48 h. However, treatment with P144 for 48 h could markedly inhibit cell invasion and migration in Lovo cells and SW480 cells. Accordingly, TGF-β1-treated Lovo cells possess a high migration and invasion ability compared with non-treated cells. Herein, we speculate that the blocking of Slit2/Robo1 signaling inhibits tumor metastasis through the targeting of TGF-β1. However, the partially abolished R5 inhibition efficacy on cell migration and invasion that occurs with TGF-β1 treatment suggests that specific blocking of Slit2/Robo1 signaling to inhibit tumor metastasis using R5 could also target other signaling pathways, at least including activation of Hakai-mediated EMT [8].

In summary, our studies have elucidated that high levels of Slit2 in tumor tissues and serum may be a potential biomarker for CRC. The blocking of Slit2/Robo1 signaling may suppress tumor metastasis by the partial inactivation of the TGF-β/Smads pathway. We speculate that inhibition of Slit2/Robo1 signaling may serve as a therapeutic method for the treatment of CRC.

## Figures and Tables

**Figure 1 cells-08-00635-f001:**
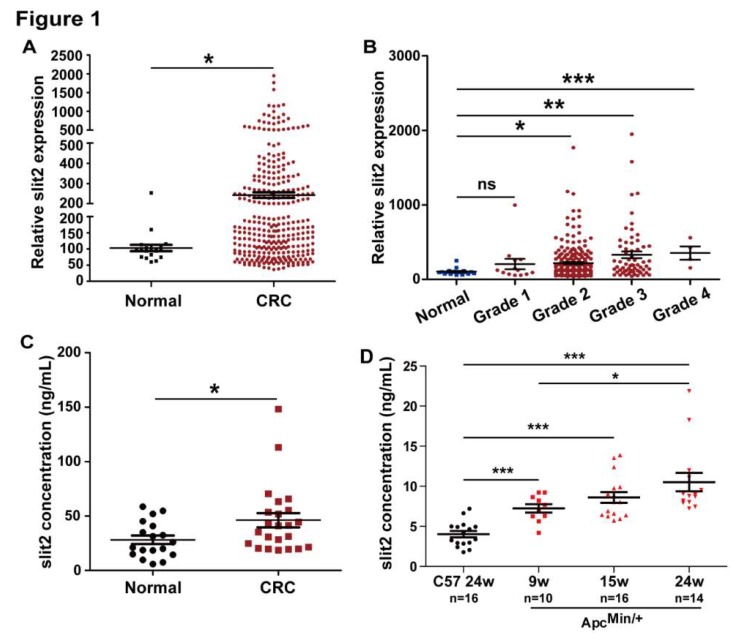
The expression of Slit2 in patients with CRC and Apc^Min/+^ mice. (**A**) Relative gene expression levels of Slit2 in the normal and primary tumor tissues of CRC cases from the MERAV database was analyzed. Relative Slit2 expression was significantly increased in the primary tumor tissues of CRC patients (*n* = 321) compared with normal samples (*n* = 19). (**B**) Relative gene expression levels of Slit2 in normal tissues and different grades of primary tumor tissues of CRC cases from the MERAV database. Relative Slit2 expression is gradually unregulated with the increase of the pathological grade of primary CRC patients. There is no statistical difference in the Slit2 expression, as shown by the *t*-test between the normal group (*n* = 19) and grade 1 primary CRC patients (*n* = 13). A significant increase of Slit2 expression can be seen between the normal group and grade 2 (*n* = 241), grade 3 (*n* = 63), and grade 4 (*n* = 4) primary CRC patients. (**C**) The protein concentration of Slit2 in the serum of healthy volunteers and CRC patients was detected by the ELISA assay. Increasing expression of Slit2 in the serum of CRC patients (*n* = 23) can be seen compared with that in normal healthy volunteers (*n* = 18). (**D**) The protein concentration of Slit2 in wild type (C57) mice or Apc^Min/+^ mice with different pathologic stages was detected by the ELISA assay. Slit2 increases in the serum of Apc^Min/+^ mice compared with in wild type mice. Additionally, Slit2 gradually increases during tumor development. *: *p* < 0.05, **: *p* < 0.01, and ***: *p* < 0.001.

**Figure 2 cells-08-00635-f002:**
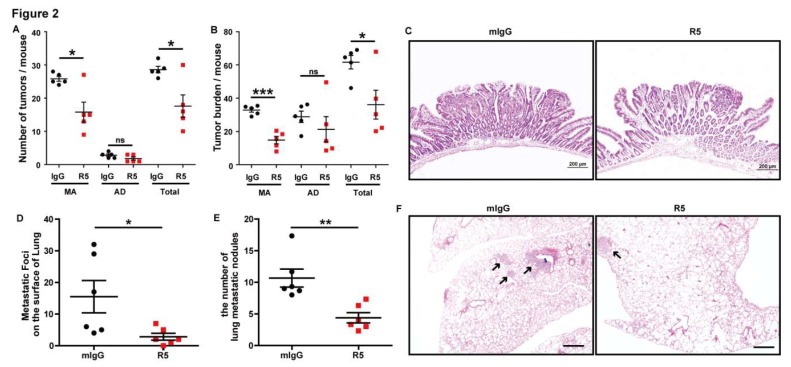
Specific blocking of Slit2/Robo1 signaling suppresses tumor growth and metastasis of CRC in vivo. The tumors were classified into microadenoma (MA, <2 mm) and adenoma (AD, ≥2 mm) only based on the tumor diameters. The number (**A**) and burden (**B**, total tumor volume in the intestine per mouse) of tumors of microadenoma and total types of tumor (both microadenoma and adenoma) were significantly inhibited by specific blockage of Slit2/Robo1 signaling by R5 compared with the mIgG group. The number and burden of adenoma were also decreased in the intestine of R5-treated Apc^Min/+^ mice compared with that in mIgG-treated mice. However, there were no significant differences in the tumor number and tumor burden between the two groups due to the individual differences of mice. (**C**) R5 treatment decreased the pathological stage of tumors in Apc^Min/+^ mice compared with that of the mIgG-treated mice. Most of the intestinal tumors were adenoma in mIgG-treated Apc^Min/+^ mice, and most hyperplasia can be seen in the intestines of R5-treated Apc^Min/+^ mice. The lung metastasis tumor model of Lovo cells was constructed and treated with R5, a monoclonal antibody that specifically recognizes human, rat, and mouse Robo1, and mIgG. The number of lung metastatic foci on the face of lung tissues was significantly decreased in the R5-treated group compared with the mIgG-treated group. (**D**) The lung tissues were cut into sections and stained with hematoxylin and eosin. The number of lung metastatic foci in lung tissue sections was counted. R5 could markedly decrease the number of metastatic foci in lung tissue sections compared with mIgG-treated mice. (**E**) Representative of metastatic foci in lung tissues sections in R5 and mIgG-treated mice. There were fewer lung metastatic foci in the slices of the R5-treated mice than in the mIgG-treated mice. (**F**) *n* = 6. *: *p* < 0.05, **: *p* < 0.01, and ***: *p* < 0.001. Scale bars: 500 μm.

**Figure 3 cells-08-00635-f003:**
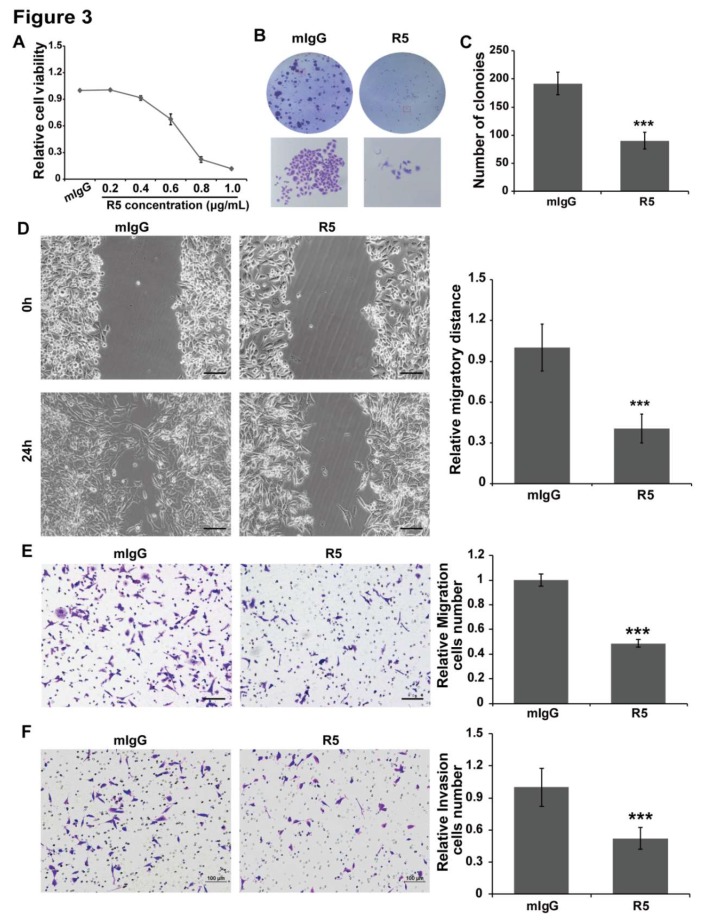
Blocking of Slit2/Robo1 signaling inhibits the growth, migration, chemotactic, and invasion ability of Lovo cells. (**A**) The indicated concentration of R5 was added to Lovo cells and the cell viability was measured using the CCK8 (Cell Counting Kit-8) assay 48 h later. R5 inhibited cell viability in a dose-dependent manner, and the IC_50_ (half maximal inhibitory concentration) is ~0.7 μg/mL in Lovo cells. Lovo cells were seeded at the density of 1000 cells and treated with 0.7 μg/mL of mIgG or R5. (**B**) The size of a single colony was smaller in R5-treated wells compared with mIgG-treated wells. (**C**) The total number of colonies in R5-treated and mIgG-treated wells of Lovo and SW480 cells were counted, and fewer colonies can be seen in R5-treated wells compared with mIgG-treated wells. (**D**) A wound healing assay was used to detect the inhibition effect of the blocking of Slit2/Robo1 signaling on cell migration. Lovo cells were treated with mIgG or R5, and the cell migration distance was detected 24 h later. R5 significantly inhibited cell migration compared with mIgG in Lovo cells. (**E**) The Transwell migration assay was used to detect the chemotactic ability of Lovo cells. The number of cells that migrated to the lower membrane was decreased in the R5 treatment group compared with that of the mIgG treatment group. (**F**) The inhibition effect on cell invasion was detected using the Matrigel invasion assay. Lovo cells were added into the upper chambers and treated with mIgG or R5 at the same time, and the invasion of the cells to the bottom of the chambers was measured 20 h later. Specific blocking of Slit2/Robo1 signaling using R5 significantly suppressed cell invasion in Lovo cells compared with mIgG. *** *p* < 0.001. Scale Bars: 100 μm. The results are representative of three independent experiments.

**Figure 4 cells-08-00635-f004:**
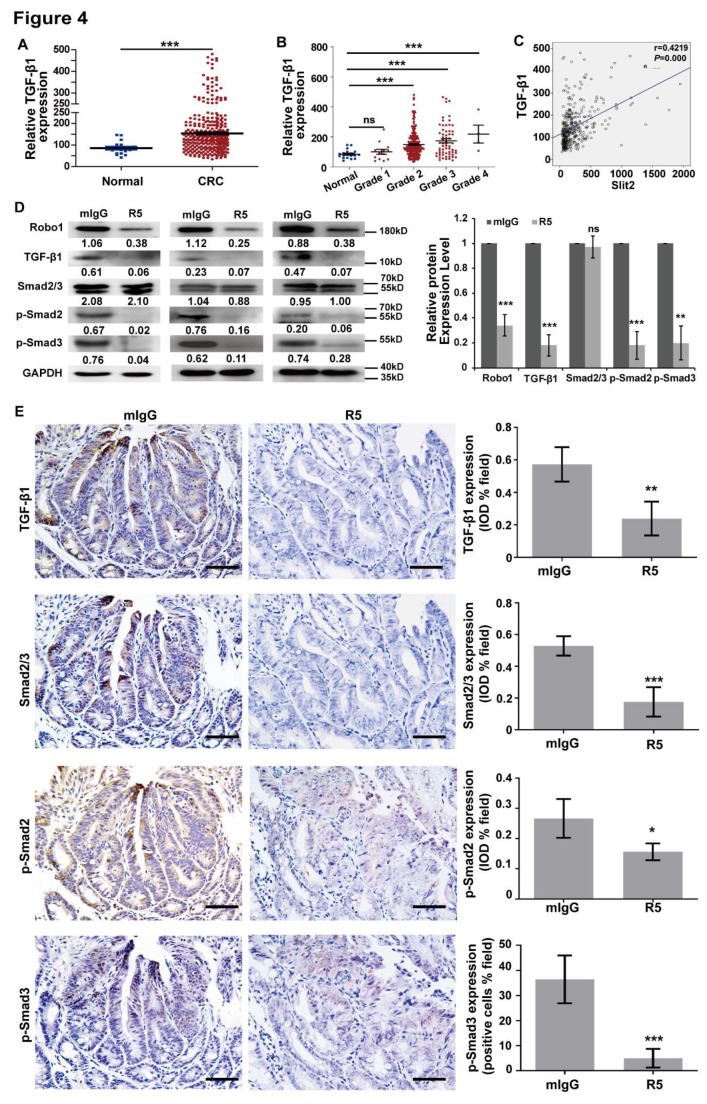
The TGF-β1/Smad2/3 pathway is correlated with Slit2/Robo1 signaling in CRC. (**A**) Relative TGF-β1 expression, taken from the MERAV database, is significantly increased in primary tumor tissues of CRC patients (*n* = 321) compared with normal samples (*n* = 19). (**B**) Relative TGF-β1 expression is gradually upregulated with the increase of the pathological grade in primary CRC patients from the MERAV database. There is no statistical difference in TGF-β1 expression, as shown by the *t*-test, between the normal group (*n* = 19) and grade 1 primary CRC patients (*n* = 13). A significant increase of TGF-β1 expression can be seen between the normal group and grade 2 (*n* = 241), grade 3 (*n* = 63), and grade 4 (*n* = 4) primary CRC patients. (**C**) The relationship between TGF-β1 and Slit2 was analyzed using the MERAV database. TGF-β1 expression was positively correlated with Slit2. (**D**) The protein expression of the TGF-β1/Smads signaling pathway in Lovo cells, which were treated with mIgG or R5, were detected by the Western blotting assay. The protein bands were quantified densitometrically using Quantity One software and normalized to GAPDH expression. The blocking of Slit2/Robo1 signaling using R5 significantly inhibits the expression of TGF-β1 and phosphorylated Smad2/Smad3 compared with in mIgG-treated Lovo cells. (**E**) The protein expression of the TGF-β1/Smads signaling pathway in tumor tissues of mIgG- or R5-treated Apc^Min/+^ mice was detected by the IHC assay. The protein expression levels in slices were quantified using IPP software. ns: no significant differentiation, *: *p* < 0.05, **: *p* < 0.01, ***: *p* < 0.001.

**Figure 5 cells-08-00635-f005:**
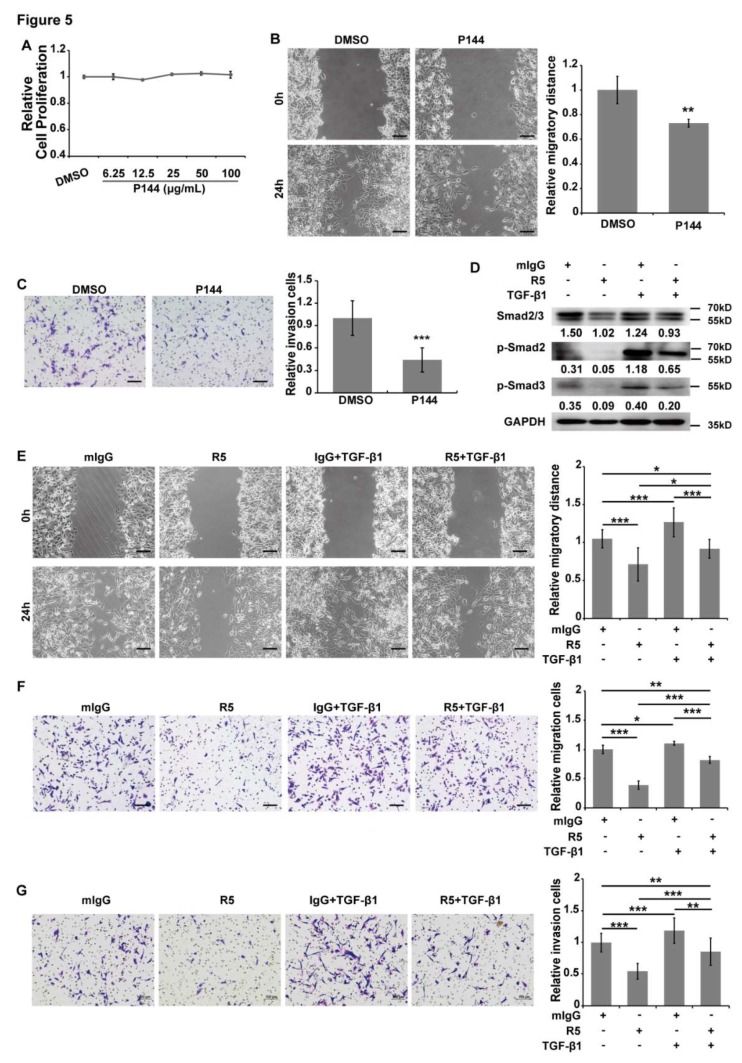
Blocking of Slit2/Robo1 signaling inhibits cell migration, chemotactic, and invasion ability, which is partially dependent on the TGF-β1/Smad2/3 pathway. (**A**) There was no significant difference in the cell proliferation of Lovo cells between the cells that were treated and those that were not treated with P144, a specific TGF-β1 inhibitor, for 48 h. Lovo cells were treated with DMSO (Dimethyl sulfoxide) or P144 and the cell migration (**B**) and invasion (**C**) abilities were detected. P144 significantly inhibits the migration and invasion ability of Lovo cells. (**D**) The inhibition of smad2 and smad3 phosphorylation by R5 was abolished by TGF-β1 treatment that was detected by the Western blotting assay. A wound healing assay (**E**), Transwell migration assay (**F**), and Matrigel invasion assay (**G**) were employed to detect whether the TGF-β1 pathway is involved in Slit2/Robo1 signaling-mediated cell migration and invasion. Lovo cells were treated with R5 or TGF-β1, or treated with both R5 and TGF-β1, to further detect cell migration, chemotactic, and invasion abilities. R5 suppressed cell migration, chemotactic, and invasion ability in Lovo cells. Inversely, treatment with TGF-β1 increased migration, chemotactic, and invasion ability in Lovo cells. Further treatment with R5 and TGF-β1 could partially reverse the inhibitory efficacy of R5 on cell migration, chemotactic, and invasion ability. *: *p* < 0.05, **: *p* < 0.01, and ***: *p* < 0.001. The protein bands were quantified densitometrically and normalized to GAPDH expression. Scale Bars: 100 μm.

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
