# Peer review of "Activation of Slit2/Robo1 Signaling Promotes Tumor Metastasis in Colorectal Carcinoma through Activation of the TGF-β/Smads Pathway"

_cells, 2019, doi:10.3390/cells8060635_

Round 1
Reviewer 1 Report
The authors should address these questions.
Clinical-pathological characteristics of the patients should be shown in detail.
Uncut gels for Western blot should be shown in supplementary figures. More replicates should be shown as well.
The proposed signaling pathway should be tested in mice with proper loss-of-function approaches.
Author Response
Response to Reviewer 1 Comments
Point 1: Clinical-pathological characteristics of the patients should be shown in detail.
Response 1: We thank this reviewer for such a meticulous proofreading and valuable suggestions. However, we have made an appointment with the CRC patients in this project that only collected serum of them to detect the expression of Slit2 and not collected other relative pathological characteristics of them. Therefore, the detail clinical-pathological characteristics of patients could not show in this manuscript.
Point 2: Uncut gels for Western blot should be shown in supplementary figures. More replicates should be shown as well.
Response 2: The uncut gels for three replicates of western blot were shown in supplementary figure 4D.
Point 3: The proposed signaling pathway should be tested in mice with proper loss-of-function approaches.
Response 3: We appreciate these constructive comments. The expression of TGF-β/Smad2/3 signaling was detected in the tumor tissues of ApcMin/+ mice, which were treated with mIgG or R5, using IHC assay. The results were added in revised figure 4E.
Reviewer 2 Report
Abstract
Line 11 define Slit2
Line 15: serum levels were significantly increased compared to what?
Line 18, “which” can be rescued, what? Cell migration and metastasis, or cell proliferation?
Line 18: the combination? Do you mean “the interaction’? Line 33-
Introduction
More references for lines 24-25 are needed
34: The concept is not clearly explained
Line 37: define for which tumor the methylation of Slit2 is not associated to stage
Line 59: cells becoming “of” an invasive phenotype..
Material and Methods
Line 64: specificLY
Line 65: reportED
Line 64 to 71: define concentrations of antibodies
Line 90: define Lovo cells
From line 91 to 94: correct the English
Line 105: correct the English
Line 113: correct the English
Line 120: which “relevant” drugs?
Results
Line 141: “..is used to ANALYZE..”
Line 164 are the wt mice the C57?if so please state that
LINE 172: DEFINE R5
Line 179-190: the description of figure 2 looks like should be in materials and methods
Line 192: “could markedly INHIBIT”
LINE 206: correct the English
Line 222: correct the English
Line 248: correct the English
Line 251: define P144
Line 259 ad 260 : correct the English
Line 269: the combined effect of TGFb and R5 still gives a reduction of invasion and migration which is significant (** and *) compared to the control treated with Igg. I can see that you argument that in discussion. Maybe you could briefly refer to that also in results
Line 303-308. The way you introduce the Wnt pathway is not clear, and not clear is also its relationship with the Slit-Robo pathway. The reference to the different treatment needs an explaination. Furthermore please correct the English
Line 336: treatment with P144 could markedly INHIBIT cell invasion and migration..
Line 341-342: correct the english
Supp Figure 2, Clarify the description of A and B
In general, what is the rationale of showing the effects of TGFb inhibition through P144 since the study is about Slit-Robo function and interaction with TGFb?
Author Response
Response to Reviewer 2 Comments
Response: We thank this reviewer for such a meticulous proofreading and valuable suggestions. The manuscript has been rigorously revised by an experienced scientist and all discernable language errors have been corrected. A point to point response to reviewer is as follows:
Abstract
Point 1: Line 11 define Slit2
Response 1: The Slit2 is defined as “Slit2 (slit guidance ligand 2) gene located at chromosome 4p15.2 and encodes a secreted glycoprotein, which is the endogenous ligand of Roundabout1 (Robo1) transmembrane receptor.” and added in the Line 11 of revised manuscript
Point 2: Line 15: serum levels were significantly increased compared to what?
Response 2: “Serum levels of Slit2 in patients with CRC or ApcMin/+ spontaneous intestinal adenoma mice were significantly increased compared with their healthy controls (healthy volunteers or wild type mice).” and have been added in the Line 15 of revised manuscript.
Point 3: Line 18, “which” can be rescued, what? Cell migration and metastasis, or cell proliferation?
Response 3: “Cell migration and metastasis can be rescued by treatment with TGF-β1.” The accurate description was added in the Line 19 of revised manuscript.
Point 4: Line 18: the combination? Do you mean “the interaction’?
Response 4: We thank the reviewer for pointing out the inaccuracy. In the manuscript, we want to describe that “the interaction of Robo1 and Slit2…” in the Lin 22 of revised manuscript.
Introduction
Point 5: More references for lines 24-25 are needed
Response 5: As per the comment, more references “list as references 2 and 3”were added in the first paragraph of revised manuscript.
Point 6: Line 34: The concept is not clearly explained
Response 6: We apologize for the poor description in the original manuscript. The description was modified as “Slit glycoprotein (Slit1, 2 and 3) can bind to the Roundabout receptor (Robo1, 2, 3 and 4) and activation of Slit/Robo signaling, which plays an important role in the cell migration that involved in physiological and pathological processes.” in Line 36-38 of the revised manuscript.
Point 7: Line 37: define for which tumor the methylation of Slit2 is not associated to stage
Response 7: The methylation of Slit2 does not associate with the clinic pathological stage in Wilms’ tumor and renal cell carcinoma. The tumor types, in which the methylation of Slit2 is not associated to pathological stage, were added in Line 42 of revised manuscript.
Point 8: Line 59: cells becoming “of” an invasive phenotype.
Response 8: We thank the reviewer for pointing out the inaccuracy. We further modified it in Line 64 of revised manuscript.
Material and Methods
Point 9: Line 64: specificLY; Line 65: reported
Response 9: These inaccuracies describe were modified in Line 70 and Line 71 of revised manuscript.
Point 10: Line 64 to 71: define concentrations of antibodies
Response 10: The concentrations of antibodies that used in IHC or WB were added in Line 74 to Line 81 of revised manuscript.
Point 11: Line 90: define Lovo cells
Response 11: The cells were defined as “Lovo human colorectal carcinoma cells” in Line 102 and “Human colorectal carcinoma cell lines, Lovo (highly invasive) and SW480 (poorly invasive), which are high expression of Slit 2” in Line 110 of revised manuscript.
Point 12: From line 91 to 94: correct the English, Line 105: correct the English, Line 113: correct the English
Response 12: All discernable language errors have been corrected by an experienced scientist
Point 13: Line 120: which “relevant” drugs?
Response 13: The relevant drugs was replaced with “mixed with the mIgG or R5 in serum free DMEM” in Line 130 of revised manuscript.
Results
Point 14: Line 141: “..is used to ANALYZE..”
Response 14: The inaccurate description has been corrected in Line 171 of revised manuscript.
Point 15: Line 164 are the wt mice the C57?if so please state that
Response 15: The ApcMin/+ mice were C57BL/6J background. Therefore, the wild type control mice were C57BL/6J, and the state was added in the revised manuscript.
Point 16: LINE 172: DEFINE R5
Response 16: R5 is defined as “R5, a monoclonal antibody that specifically recognizes human, rat and mouse Robo1, could inhibit Robo1 expression [8,18]” and added in Line 207 of revised manuscript.
Point 17: Line 179-190: the description of figure 2 looks like should be in materials and methods
Response 17: The detailed description of figure legend was removed to materials and methods.
Point 18: Line 192: “could markedly INHIBIT”; LINE 206: correct the English; Line 222: correct the English; Line 248: correct the English
Response 18: All discernable language errors have been corrected by an experienced scientist
Point 19: Line 251: define P144
Response 19: P144 is defined as “P144 is a synthetic peptide that specific inhibit of TGF-β1 and often used to inactivate TGF-β/Smad signaling” and was added in Line 317 of revised manuscript.
Point 20: Line 259 and 260: correct the English
Response 20: As per reviewer’s suggestion, the language of manuscript has been modified by an experienced scientist.
Point 21: Line 269: the combined effect of TGFb and R5 still gives a reduction of invasion and migration which is significant (** and *) compared to the control treated with Igg. I can see that you argument that in discussion. Maybe you could briefly refer to that also in results
Response 21: We appreciate the constructive comment. The rescue data was described as “In comparison, the inhibition of Smad2 and Smad3 phosphorylation by R5 can be absolutely abolished by TGF-β1 treatment (Figure 5D). Unexpectedly, the inhibition efficacy of R5 on cell migration, chemotactic and invasion of Lovo cells can be partially abolished by treated with TGF-β1 (Figure 5E-5G). It was implied that TGF-β/Smad may be not the only signaling pathway that involving in Slit2/Robo1-mediated tumor metastasis in CRC.” and added in revised “results 3.5”.
Point 22: Line 303-308. The way you introduce the Wnt pathway is not clear, and not clear is also its relationship with the Slit-Robo pathway. The reference to the different treatment needs an explaination. Furthermore please correct the English
Response 22: We apologize for the confusion. The Wnt pathway is not associated with this manuscript. I forgot to delete the introduction about Wnt pathway when prepared this manuscript. Therefore, this part description for Wnt pathway was deleted in revised manuscript. In addition, the language of manuscript has been modified by an experienced scientist.
Point 23: Line 336: treatment with P144 could markedly INHIBIT cell invasion and migration..; Line 341-342: correct the English
Response 23: This inaccurate description has been corrected as “treatment with P144 for 48 h could markedly inhibit cell invasion and migration in Lovo cells and SW480 cells.” The language of manuscript has been modified by an experienced scientist.
Point 24: Supp Figure 2, Clarify the description of A and B
Response 24: As per reviewers’ suggestion, the figures in the main text and supplementary were rearranged in revised manuscript. The detailed description was added in the Figure legends of the revised manuscript.
Point 25: In general, what is the rationale of showing the effects of TGFb inhibition through P144 since the study is about Slit-Robo function and interaction with TGFb?
Response 25: We using P144 to specific inhibit TGF-β1 and further detect the role of TGF-β1 on cell proliferation, migration and invasion in CRC cells. We try to clarify the TGF-β-dependent function of Slit2/Robo1 in CRC. However, we have deleted the verification western blotting results of P144 on the inhibition of TGF-β1/Smads signaling as per reviewer’s suggestion
Reviewer 3 Report
In this manuscript the authors analyse the participation of Slit2/Robo1 signalling in CRC progression and a possible connection with TGFb/Smad pathway. The study is well designed and the findings potentially relevant to signalling in colorectal cancer cells and CRC metastasis. However, I have some comments to some of the data presented:
1- It would be useful to remind in the text the mechanism of action of the R5 antibody to block Robo1 activation. Same for action of P144 on TGFb
2- Is it correct the definition of microadenoma and adenoma in Supplementary Figure 2? Is 2um the correct length? The explanation of the difference between the two is confusing and it is rather important for the interpretation of the results. The images do not help on this matter. This should be clarified and the functional difference between the two explained.
3- I have some doubts about the wound-healing assay being the proper way to address an effect on cell migration in this setting. This assay is sensitive to cell proliferation and the authors describe an effect of R5 on CRC cell proliferation. This could greatly affect the assay. Moreover, images at Time 0 in wound-healing assays show that the morphology of the cells after R5 treatment could be also different, affecting also the results. The best way to address this issue is to perform other migration assays, like live cell tracking or chemotaxis assays, and to check for changes in cell shape or adhesion.
4- Since authors mention in a previous paper a TGFb independent effect of Robo1 on Smad2/3, it would be good to perform rescue experiments to further dilucidated if the effect on phosphoSmads is due to the drop in TGFb after R5 treatment.
5- I am not fully convinced of the interpretation of the data from Figure5. As TGFb increases cell migration and invasion, the partial rescue after R5 treatment does not necessarily mean that they are on the same pathway. The title on section 3.5 is an overstatement. Even more when no metastasis assays have been performed to demonstrate that.
Minor points:
- In my opinion, some supplementary figures belong to the main text, like supplementary Figure 2.
- English should be improved throughout the manuscript.
- Use correct translation of EMT in the text.
Author Response
Response to Reviewer 3 Comments
Point 1: It would be useful to remind in the text the mechanism of action of the R5 antibody to block Robo1 activation. Same for action of P144 on TGFb
Response 1: R5 is a monoclonal antibody that specific recognizes Robo1 and could inhibit the expression of Robo1. P144 (Disitertide) is a synthetic peptide that specific inhibit of TGF-β1 and often used to inactivate TGF-β/Smad signaling. The relative references about the action mechanism of R5 and P144 were added in the revised manuscript.
Point 2: Is it correct the definition of microadenoma and adenoma in Supplementary Figure 2? Is 2um the correct length? The explanation of the difference between the two is confusing and it is rather important for the interpretation of the results. The images do not help on this matter. This should be clarified and the functional difference between the two explained.
Response 2: We appreciate these constructive comments. It was our typing errors to define of microadenoma and adenoma. The tumor should be defined base on the tumor diameters, and the tumor was categorized as microadenoma (<2 mm) and adenoma (≥2 mm) according previously report [1]. The ApcMin/+ mouse is an intestinal tumor model that spontaneously develop multiple adenomatous polyps in the small intestine and fewer polyps in the colon. There are no functional differences between microadenoma and adenoma. It is only the size different between the two types of tumors. The number of microadenomas were higher than that of adenomas in ApcMin/+ mice. In our data, very few numbers of adenomas (≥2 mm) can be seen in 11-week old ApcMin/+ mice. The tumor number and tumor volume of adenomas were decreased in R5 treated ApcMin/+ mice compared with the mIgG treatment mice. Unfortunately, the statistical analysis results showed no significantly differences due to the individual differences of one of the mice that treated with R5. Therefore, we still believed that specific inhibition of Slit2/Robo1 signalling by R5 can decrease the growth of intestinal tumors. The raw data were given in supplementary table1. If the results are still confused for readers, it could statistical analyse the total tumor number and tumor burden that do not consider the tumor diameters.
Reference: 1. Qi, C.; Li, B.; Guo, S.; Wei, B.; Shao, C.; Li, J.; Yang, Y.; Zhang, Q.; Li, J.; He, X., et al. P-Selectin-Mediated Adhesion between Platelets and Tumor Cells Promotes Intestinal Tumorigenesis in Apc(Min/+) Mice. International journal of biological sciences 2015, 11, 679-687, doi:10.7150/ijbs.11589.
Point 3: I have some doubts about the wound-healing assay being the proper way to address an effect on cell migration in this setting. This assay is sensitive to cell proliferation and the authors describe an effect of R5 on CRC cell proliferation. This could greatly affect the assay. Moreover, images at Time 0 in wound-healing assays show that the morphology of the cells after R5 treatment could be also different, affecting also the results. The best way to address this issue is to perform other migration assays, like live cell tracking or chemotaxis assays, and to check for changes in cell shape or adhesion.
Response 3: As per reviewer’s suggestion, we further detect cell chemotactic capacity of cells using transwell migration assay. The results were added in revised figure 3 and revised figure 5.
Point 4: Since authors mention in a previous paper a TGFb independent effect of Robo1 on Smad2/3, it would be good to perform rescue experiments to further dilucidated if the effect on phosphoSmads is due to the drop in TGFb after R5 treatment.
Response 4: The rescue experiments were carried out and further detected using immunoblotting assay. We found that R5 significantly inhibit the expression of TGF-β1 and phosphorylation of Smad2/3 compared with in mIgG-treated group in Lovo cells. However, the phosphorylation of Smad2/3 cannot be further inhibited by R5 while simultaneously treated with TGF-β1 in Lovo cells. The results were added in revised figure 5.
Point 5: I am not fully convinced of the interpretation of the data from Figure5. As TGFb increases cell migration and invasion, the partial rescue after R5 treatment does not necessarily mean that they are on the same pathway. The title on section 3.5 is an overstatement. Even more when no metastasis assays have been performed to demonstrate that.
Response 5: We further confirmed that R5 inhibit the expression of TGF-β1 and phosphorylation of Smad2/3 compared with in mIgG-treated group in Lovo cells. However, the phosphorylation of Smad2/3 cannot be further inhibited by R5 while simultaneously treated with TGF-β1 in Lovo cells. These results can certificate that Slit2/Robo1 could regulate TGF-β1/Smad2/3 signaling pathway. The previously report indicated that Slit2/Robo1 also regulated Hakai-mediated cell migration and invasion in colorectal carcinoma. Therefore, the partial rescue after R5 treatment implied that Slit2/Robo1-mediated tumor metastasis not only by regulate TGF-β1/Smad2/3 signaling pathway, but also by regulate Hakai or other unclear mechanism. In addition, we confirmed that specific blockage of Slit2/Robo1 by R5 could inhibit cell proliferation, migration and invasion of Lovo cells and SW480 cells. However, P144 was used to inhibit TGF-β1/Smads signaling and found that inactivation of TGF-β1/Smads signaling can significantly inhibit cell migration and invasion but not suppressing cell proliferation in tumor cells of CRC. These results also indicated that TGF-β1/Smads signaling is involved in Slit2/Robo1-mediated tumor metastasis.
Minor points:
Point 6: In my opinion, some supplementary figures belong to the main text, like supplementary Figure 2.
Response 6: We appreciate these constructive comments. The figures were rearranged in revised manuscript.
Point 7: English should be improved throughout the manuscript.
Response 7: The language of manuscript has been modified by an experienced scientist.
Point 8: Use correct translation of EMT in the text.
Response 8: The translation of EMT was corrected in Line 64 of revised manuscript.
Round 2
Reviewer 3 Report
English should still be edited, at least on abstract. Very difficult to read.
Author Response
Response to Reviewer 3 Comments
Point 1: English should still be edited, at least on abstract. Very difficult to read.
Response 1: The abstract in the revised manuscript has been further modified by an experienced scientist.